# The human RNA polymerase I structure reveals an HMG-like docking domain specific to metazoans

Julia L Daiß[1], Michael Pilsl[1], Kristina Straub[1], Andrea Bleckmann[1], Mona Höcherl[1], Florian B Heiss[1], Guillermo Abascal-Palacios[2,3,4], Ewan P Ramsay[2,5], Katarina Tlučková[6], Jean-Clement Mars[7,8,9], Torben Fürtges[10], Astrid Bruckmann[1], Till Rudack[10], Carrie Bernecky[6], Valérie Lamour[11,12], Konstantin Panov[13], Alessandro Vannini[2,5], Tom Moss[7,8], Christoph Engel[1]

Transcription of the ribosomal RNA precursor by RNA polymerase (Pol) I is a major determinant of cellular growth, and dysregulation is observed in many cancer types. Here, we present the purification of human Pol I from cells carrying a genomic GFP fusion on the largest subunit allowing the structural and functional analysis of the enzyme across species. In contrast to yeast, human Pol I carries a single-subunit stalk, and in vitro transcription indicates a reduced proofreading activity. Determination of the human Pol I cryo-EM reconstruction in a close-to-native state rationalizes the effects of disease-associated mutations and uncovers an additional domain that is built into the sequence of Pol I subunit RPA1. This "dock II" domain resembles a truncated HMG box incapable of DNA binding which may serve as a downstream transcription factor–binding platform in metazoans. Biochemical analysis, in situ modelling, and ChIP data indicate that Topoisomerase 2a can be recruited to Pol I via the domain and cooperates with the HMG box domain–containing factor UBF. These adaptations of the metazoan Pol I transcription system may allow efficient release of positive DNA supercoils accumulating downstream of the transcription bubble.

## Introduction

Transcription of DNA into RNA is carried out by three nuclear polymerases (Pols) in most higher eukaryotes (1). These multi-subunit Pols diverge in target loci, structure, and regulation (2). Understanding the underlying molecular mechanisms is a central goal of molecular biology. However, these mechanisms have been mostly studied in lower model organisms because of experimental limitations. In higher eukaryotes, regulatory variations dependent on tissue type, developmental state, and cell-cycle stage are adding additional layers of complexity (3).

Human RNA polymerase (hPol) I has a single target gene, the 47S ribosomal RNA precursor (pre-rRNA), from which the 5.8S, 18S, and 28S rRNA are processed (4). These processed RNAs contribute to ribosome formation together with the 5S rRNA synthesized by Pol III (5). rRNA synthesis contributes up to 80% of total cellular RNA (6) and must therefore be tightly regulated. Hence, dysregulation of hPol I is associated with pathologies, such as cancer and developmental diseases, for example, Treacher Collins Syndrome (7). Unsurprisingly, inhibition of hPol I has been explored as a therapeutic strategy with some success in cancer treatment and future potential (8). The molecular action of rRNA synthesis inhibitors is not entirely understood and may range from the activation of DNA-damage responses upon interference with replication (9) to a specific reduction of Pol I transcription by preventing promoter escape during initiation (10) or inhibiting elongation (11).

The composition of hPol I is similar to yeast Pol I (12) of which detailed crystal structures are known (13, 14). A catalytic core of 10 subunits is complemented by a protruding stalk subcomplex and a heterodimeric RPA49/RPA34 subcomplex. The latter is related to Pol II initiation factors TFIIF, and TFIIE (15) and has homologues in Pol III (16). The stalk was proposed to be divergent between yeast and human, as DNA- and protein sequence–based searches have not identified a homologue of subunit A14 in human cells and the lack of which was recently confirmed by structural investigations of the human enzyme (17, 18). Table 1 summarizes the subunit terminologies for yeast and mammalian Pol I in comparison with human Pol II and Pol III subunits and correlates nomenclature. The structure–function analysis of yeast and human Pol II (19) and Pol III

[1]Regensburg Center for Biochemistry, University of Regensburg, Regensburg, Germany    [2]Division of Structural Biology, The Institute of Cancer Research, London, UK    [3]Biofisika Institute (CSIC, UPV/EHU), Leioa, Spain    [4]IKERBASQUE, Basque Foundation for Science, Bilbao, Spain    [5]Fondazione Human Technopole, Structural Biology Research Centre, Milan, Italy    [6]Institute of Science and Technology, Klosterneuburg, Austria    [7]Department of Molecular Biology, Medical Biochemistry and Pathology, Faculty of Medicine, Laval University, Quebec, Canada    [8]Laboratory of Growth and Development, St-Patrick Research Group in Basic Oncology, Cancer Division of the Quebec University Hospital Research Centre, Québec, Canada    [9]Borden Laboratory, IRIC, Université de Montréal, Montréal, Québec, Canada    [10]Protein Crystallography, Department of Biophysics, Faculty of Biology and Biotechnology, Ruhr University Bochum, Bochum, Germany    [11]Université de Strasbourg, CNRS, INSERM, Institut de Génétique et de Biologie Moléculaire et Cellulaire (IGBMC), Department of Integrated Structural Biology, Illkirch, France    [12]Hôpitaux Universitaires de Strasbourg, Strasbourg, France    [13]School of Biological Sciences and PGJCCR, Queen's University Belfast, Belfast, UK

Correspondence: christoph.engel@ur.de

**Table 1.  Human Pol I, II, and III subunit nomenclature in relation to yeast counterparts.**

| Sc Pol I | | Sp Pol I | | Mm Pol I | | Hs Pol I | | Hs Pol II | | Hs Pol III | |
|---|---|---|---|---|---|---|---|---|---|---|---|
| Protein | Gene | Protein | Gene | Protein | Gene | Protein | Gene | Protein | Gene | Protein | Gene |
| A190 | RPA190 | A190 | rpa1 | RPA1 | Polr1a | RPA1 | POLR1A | RPB1 | POLR2A | RPC1 | POLR3A |
| A135 | RPA135 | A135 | rpa2 | RPA2 | Polr1b | RPA2 | POLR1B | RPB2 | POLR2B | RPC2 | POLR3B |
| AC40 | RPC40 | AC40 | rpc40 | RPAC1 | Polr1c | RPAC1 | POLR1C | RPB3 | POLR2C | RPAC1 | POLR1C |
| AC19 | RPC19 | AC19 | rpc19 | RPAC2 | Polr1d | RPAC2 | POLR1D | RPB11 | POLR2J | RPAC2 | POLR1D |
| Rpb5 | RPB5 | Rpb5 | rpb5 | RPABC1 | Polr2e | RPABC1 | POLR2E | RPABC1 | POLR2E | RPABC1 | POLR2E |
| Rpb6 | RPO26 | Rpb6 | rpb6 | RPABC2 | Polr2f | RPABC2 | POLR2F | RPABC2 | POLR2F | RPABC2 | POLR2F |
| Rpb8 | RPB8 | Rpb8 | rpb8 | RPABC3 | Polr2h | RPABC3 | POLR2H | RPABC3 | POLR2H | RPABC3 | POLR2H |
| Rbp10 | RPB10 | Rbp10 | rpb10 | RPABC5 | Polr2l | RPABC5 | POLR2L | RPABC5 | POLR2L | RPABC5 | POLR2L |
| Rbp12 | RPC10 | Rbp12 | rpc10 | RPABC4 | Polr2k | RPABC4 | POLR2K | RPABC4 | POLR2K | RPABC4 | POLR2K |
| A14 | RPA14 | A14 | ker1 | - | - | - | - | RPB4 | POLR2D | RPC9 | CRCP |
| A43 | RPA43 | A43 | rpa43 | RPA43 | Polr1f (Twistnb) | RPA43 | POLR1F (TWISTNB | RPB7 | POLR2G | RPC8 | POLR3H |
| A12.2 | RPA12 | A12.2 | rpa12 | RPA12 | Polr1h (Zndr1) | RPA12 | POLR1H (ZNDR1) | RPB9 | POLR2I | RPC10 | POLR3K |
| A49 | RPA49 | A49 | rpa49 | RPA49 (PAF53) | Polr1e (Paf53) | RPA49 | POLR1E (PAF53) | | | RPC5 | POLR3E |
| A34.5 | RPA34 | A34.5 | rpa34 | RPA34 (PAF49) | Polr1g (Paf49 / Cd3eap / Ase1) | RPA34 | POLR1G (CD3EAP / CAST / PAF49 / ASE1) | | | RPC4 | POLR3D |
| | | | | | | | | | | RPC3 | POLR3C |
| | | | | | | | | | | RPC6 | POLR3F |
| | | | | | | | | | | RPC7 | POLR3G |

Wheat: large subunits; green: subunits shared between Pol I and III; grey: common subunits; blue: stalk subcomplex; orange: Rpb9/TFIIS-like subunits; pink: built-in TFIIF/E-like subunits of Pol I and III.

(20, 21, 22, 23) showed both similarities in the catalytic mechanisms and divergence in regulatory elements among organisms. Recently published Pol I elongation complex structures showed an increased flexibility of the clamp domain within the human enzyme with additional clamp–DNA contacts present in elongation complex structures, whereas the clamp domain was open and showed increased flexibility in an inactive state (17, 18). Furthermore, the stalk subcomplex of human Pol I also shows increased flexibility because of a reduced number of contacts with the Pol I core but shows significant movement upon interaction with initiation factor Rrn3 (18).

The factor Rrn3 itself is essentially conserved among species (24, 25, 26) and primes Pol I for initiation by interacting with the stalk subcomplex (27, 28, 29, 30). Regulation of Pol I is diverse (31) and can be achieved by post-translational modification of Pol I subunits or transcription factors. Nutrient availability (32) and growth factor signal transduction (33) activate Pol I initiation by (de-)phosphorylation of initiation factor Rrn3. Dephosphorylation of the stalk is required for efficient Pol I function in yeast (34), and hyper-acetylation of RPA49 reduces Pol I activity under stress (35).

Apparently, many factors of the Pol I transcription system are conserved functionally but diverge in composition (36). In addition to RRN3, hPol I transcription requires the initiation factors "Selectivity Factor 1" (SL1) and "upstream binding factor" (UBF). SL1 comprises the subunits TAF1A, TAF1B, TAF1C (homologues of yeast Core Factor subunits), and the two additional factors TAF1D (37) and TAF12 (38), and includes the TATA-binding protein (TBP). UBF consists of six consecutive HMG boxes, is a part of initiation complexes (39) in one splice variant (40), and binds to the body of actively transcribed rDNA genes (41), apparently preventing re-association of nucleosomes.

Functionally, hPol I transcription has been studied in extracts or partially purified systems (42). In contrast, yeast Pol I transcription was studied in detail using purified and recombinantly expressed components, allowing a clear definition of subunit functionalities in transcription initiation (28, 43), elongation (44, 45), cleavage (46), backtracking (47, 48), and termination (49, 50). Such studies allowed a detailed dissection of (sub-)domain and transcription factor functions.

Because of the lack of a well-defined human in vitro system consisting of purified components, it is unclear whether the results of structure–function studies can be easily transferred to higher organisms. Hence, it remains poorly understood how Pol I structurally and functionally adapted to the increased regulatory demands in human cells, even though the first structures of hPol I started to shed some light on the matter (17, 18). Here, we show how hPol I can be exclusively purified from a modified human cell line in its natural form and determine the structure of its non-crosslinked apo form by single-particle electron cryo-microscopy (cryo-EM). Strikingly, our structure reveals a previously unknown, built-in platform that may allow docking of transcription factors on the downstream face of the polymerase. Detailed phylogenetic analysis allows tracking Pol I domain evolution including the loss of a subunit and the gain of additional domains in higher organisms. In vitro functional analysis finally demonstrates reduced proofreading ability of the human enzyme and structural mapping of known mutations give insights into the molecular basis of Pol I-related pathologies. With this, our study completes mammalian Pol I domain definitions, provides a phylogenetic analysis in context of evolving transcription factors, and demonstrates functional differences of Pol I function among species in vitro, while confirming recent structural analyses of crosslinked complexes in a more native setting.

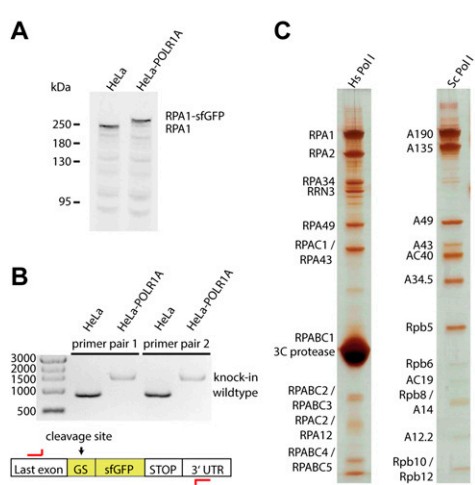
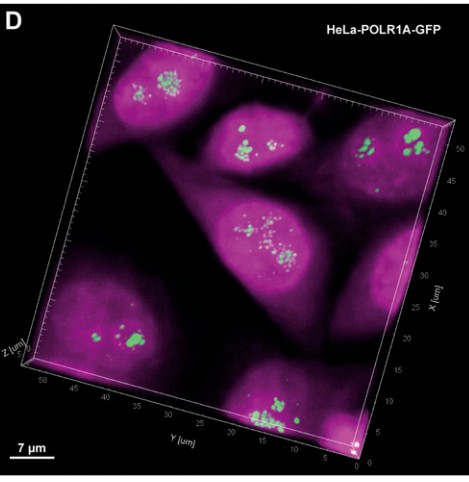

**Figure 1. Homozygous sfGFP knock-in cell line generation and human Pol I purification.**
**(A)** Western blot against RPA1 shows a shift to larger molecular weight in lysates of the POLR1A-sfGFP cell line, confirming exclusive expression of the modified protein. **(B)** Site-specific knock-in of the cleavable sfGPF fusion confirmed by PCR from genomic DNA (Sybr-Safe stained agarose gel). **(C)** Purification of human Pol I shows bands for all subunits in comparison to the *Saccharomyces cerevisiae* enzyme (silver-stained SDS–PAGE). **(D)** Confocal imaging shows the exclusive location of GFP-induced fluorescence in the nucleoli in aligned 3D stacks. Spots in the central cell may represent single rDNA genes. Magenta: DAPI stain; Green: sfGFP signal (fused to RPA1).

# Results

### Specific tagging and purification of human RNA polymerase I

To study the structure and function of hPol I in vitro, we first created a cell line that allows the specific enrichment of the complete enzyme in its native state without contamination of hPol III. Using the CRISPR/Cas9 technology in a dual-nicking approach, a cleavable sfGFP tag was fused to the genomic sequence of the largest Pol I subunit RPA1 of the Hela P2 cell line (51). After identification of positive clones by single-cell FACS based on GFP fluorescence intensity, correct insertion was confirmed by site-specific PCR. Homozygous insertion was verified by Western blot against subunit RPA1 (Fig 1). The approach we previously reported for the generation of an RPAC1-tagged cell line (20) can hence be generally applied for reliable homozygous knock-in of the C-terminal fusion tags.

hPol I purification from lysates of the RPA1-sfGFP cell line relies on a single affinity purification step followed by site-specific tag-cleavage, resulting in a highly enriched sample (Figs 1C and S1). As judged by mass spectrometry (Fig S2), the sample partially co-purifies with the initiation factor RRN3 and contains stoichiometric amounts of hPol I subunits, including the RPA49/RPA34 subcomplex, which is sub-stoichiometric in rat Pol I purifications (52). An optional subsequent ion-exchange chromatography step resulted in the loss of initiation factor RRN3 and the RPA49/34 subcomplex from most polymerases (Fig S1B).

### Human Pol I shows reduced proofreading in vitro

Equipped with a cell line that allows the specific enrichment of hPol I, we now aimed at a detailed structural and functional characterization of this enzyme in vitro. To understand functional conservation, we first compared purified hPol I activity with its counterparts from *Saccharomyces cerevisiae* and *Schizosaccharomyces pombe* in an in vitro elongation and cleavage assay. A fluorescently labeled RNA primer is extended in the presence of nucleotide triphosphates (NTPs) by Pol I, or cleaved because of the

action of the TFIIS-related subunit RPA12 (Fig 2A). Whereas yeast Pol I specifically incorporates the correctly base-paired substrate, hPol I generates transcripts containing incorrectly incorporated NTPs under identical experimental conditions (Fig 2B).

Furthermore, the cleavage pattern of yeast and human Pol I in the absence of NTPs diverges. Whereas the 3′-end of the perfectly base-paired RNA primer can be cleaved up to three nucleotides by Sc and Sp Pol I, the main product of hPol I cleavage is at position-1, indicating a reduced backtracking ability. To exclude effects from potential sub-stoichiometry of the RPA49/34 complex, we added recombinant co-expressed human RPA49/34, but observed neither increased backtracking/cleavage, nor reduced generation of mismatched transcripts (Fig S3B). Similarly, the addition of recombinant Rrn3 to Sc Pol I does not hamper its functionality (Fig S3D), suggesting that the observed effects do not originate from RRN3 present in the sample.

To test the influence of the substrate scaffold, we added a non-template (nt) strand with a mismatched bubble and tested a wealth of different template sequences (Fig S3D–I). On a mismatched bubble-template, backtracking is impaired even further, whereas the incorporation of incorrect NTPs generally remained, but showed some sequence specific variations in intensity.

Recent functional analysis indicated that yeast Pol I is more promiscuous in single nucleotide incorporation compared with Pol II in vitro (53). Our results may indicate that such an effect is even more pronounced in human Pol I, possibly originating from the flexibility among Pol I core and shelf modules as discussed (54, 55). To understand the evolution of Pol I and to rationalize the functional differences between the enzymes of different species, we determined the structure of human Pol I by cryo-EM.

### Structure determination of hPol I in its apo form

Whereas the structure of yeast Pol I has been extensively studied by X-ray crystallography (13, 14, 56) and single-particle cryo-EM (57, 58, 59), the human enzyme eluded structural characterization until recently (17, 18). In a first step, negative stain EM screening

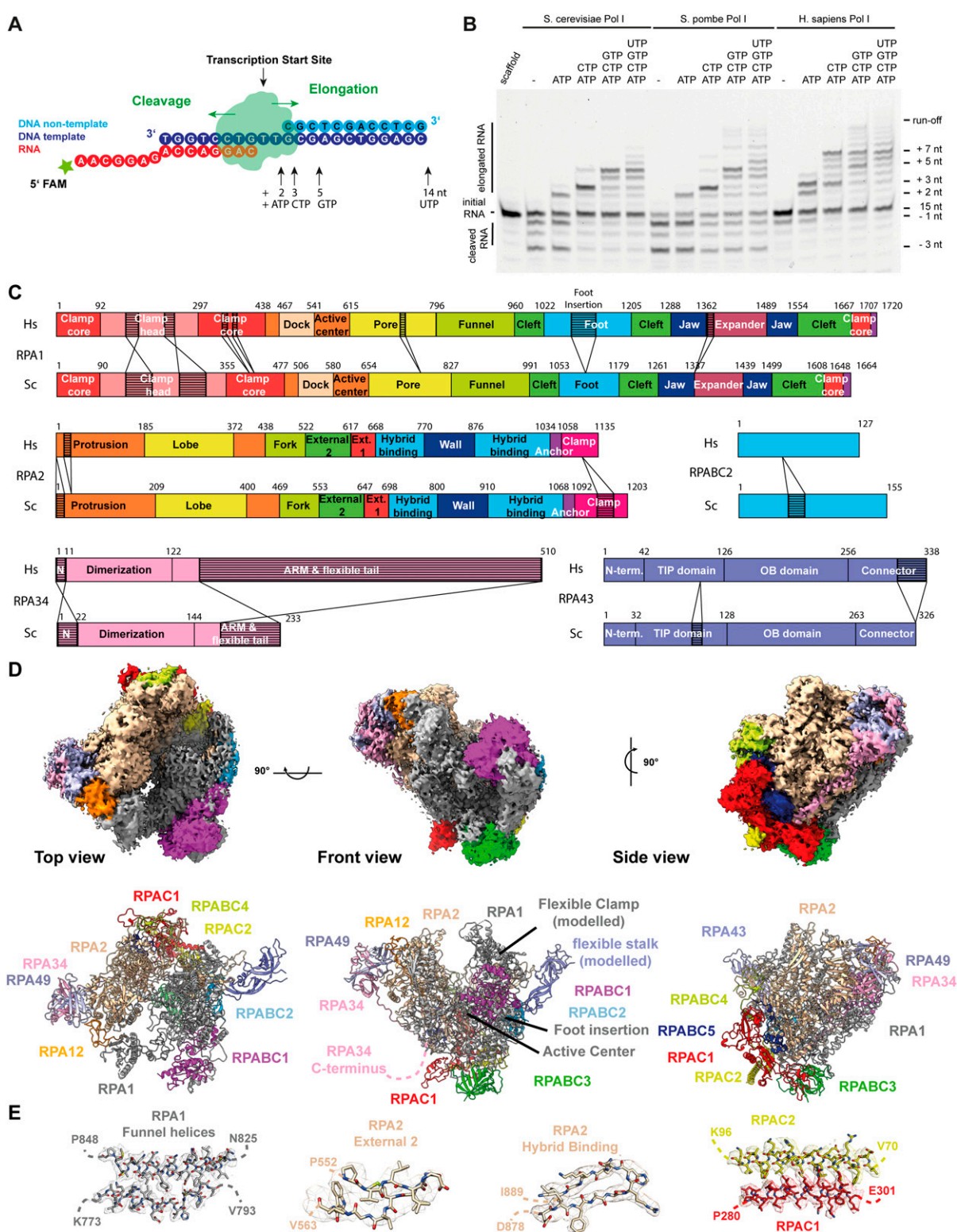

**Figure 2. Activity, domain architecture and cryo-EM reconstruction of human Pol I.**
**(A)** Schematic representation of the assays and scaffold sequences used to determine hPol I activity in vitro. **(B)** Compared with Sc and Sp Pol I, cleavage activity of hPol I is reduced and only reaches the −1 position, whereas Sc and Sp enzymes can cleave up to three nucleotides from a matched hybrid. Elongation efficiency is comparable, although incorporation of mis-matched nucleotides is strongly increased in the case of hPol I. **(C)** Schematic domain architecture of the Pol I subunits with largest differences to their yeast Pol I counterparts: RPA1, RPA2, RPA34, RPA43, and RPABC2. Subdomains and insertions/deletions of 10 or more residues indicated. **(D)** Cryo-EM density of human Pol I shows flexibilities in the clamp/stalk region of RPA1 and RPA43. Structure model shown below. **(E)** Enlarged view of RPA1 funnel helices, RPA2 External II and Hybrid Binding domains, and the RPAC1/2 assembly overlaid with sharpened cryo-EM density.

revealed intact particles (Fig S1C–E) and a 3D reconstructed neg-
ative stain envelope indicated an architecture comparable to *S.
cerevisiae* Pol I. However, most particles show flexibilities in the
clamp/stalk region that originate from heterogeneity or functional
flexibility, which is in line with the structures of elongating hPol I
and its open complex (17, 18). High-resolution structure determi-
nation by single-particle cryo-EM was hampered by intrinsic flex-
ibility and a strong bias in orientation distribution of hPol I
particles. Finally, data collected from self-made graphene oxide–
covered grids reduced orientational bias of non-crosslinked par-
ticles after extensive screening for preparation conditions (60). We
collected a total of 9,709 micrograph movies on a CryoARM 200
(JEOL) electron microscope equipped with K2 direct electron de-
tector (Gatan) at a pixel size of 0.968. Preprocessing and particle
picking in Warp (61) was followed by binning and 2D classification in
RELION 4.0 (62), yielding 145,554 particles that were subsequently
subjected to sequential 3D classification (Fig S4 and Table S1). A 3D
reconstruction with an overall resolution of 4.09 was obtained,
revealing secondary structures for most regions of the molecule.
Models for common subunits RPABC1-ABC5 and the RPAC1/2 as-
sembly were transferred from a hPol III reconstruction (21) as a
starting point for density fitting. Homology models of the hPol I
subunits RPA1, RPA2, RPA49, RPA34, RPA12, and RPA43 were gen-
erated based on sequence and secondary structure alignments
with the crystal structures of their *S. cerevisiae* counterparts
(Supplemental Data 1) using the MODELLER software package (63).
Model fitting and rigid body refinement allowed interpretation of
both negative stain and cryo-EM densities and later supported by
AlphaFold predictions (64).

To the knowledge of the authors, this is the first example for the
de novo reconstruction of a previously unknown, non-symmetric
macromolecule obtained with a CryoARM 200 electron microscope.
Details of data collection and handling strategies are similar to
recent reports (65, 66, 67) and are described in the Materials and
Methods section.

## Insights into hPol I architecture

The negative stain density shows the RPA49/RPA34 heterodimer in
most polymerases (Fig S1E). However, some 3D classes lack density
for the region of the clamp core and clamp head domain of subunit
RPA1 and the stalk subcomplex, indicating a high flexibility of this
sub-assembly.

The cryo-EM reconstruction of hPol I (Fig 2D) shows connected
density for the common hPol subunits RPABC1-5, the RPAC1/2 di-
mer, the N-terminal domain of subunit RPA12, and most parts of
subunit RPA2, with exception of the C-terminal clamp and anchor
domains (residues 1,010–1,134). Furthermore, density for the jaw,
funnel, foot, and most parts of the cleft domain of subunit RPA1
(residues 630–1,661 excluding loops) and for the RPA49/34 heter-
odimer allowed unambiguous fitting of homology models. Similar
to yeast Pol I crystal structures and cryo-EM reconstructions of apo
Pol I, the linker and tWH domains of subunit RPA49 are flexible in
human Pol I. The N terminus of subunit RPA12 can be placed on
the lobe of subunit RPA2, demonstrating stable association of the
subunit. A C-terminal extension of subunit RPA34 specific to the
human enzyme is flexible in the cryo-EM reconstruction whereas

the assembly of RPAC1/2 reflects the conformation known in hPol III
and tightly interacts with subunit RPA2. In our reconstruction, weak
density for the stalk subcomplex, the clamp and dock domains of
subunit RPA1 indicate increased shelf module flexibility but could
also originate from technical drawbacks of freezing or sample
preparation.

Global contraction of Pol I modules upon activation has been
observed in the enzymes of *S. cerevisiae* (13, 14) and *S. pombe* (68)
and may also be a regulatory feature of hPol I (54, 55) Overall, the
architecture of hPol I reflects that of the yeast counterparts, but
allows more detailed insights into the effects of Pol I-related
mutations identified in human disease and reveals two major
adaptations accumulating upon evolution: the stalk subcomplex
(flexible in our density) and the RPA1 foot domain.

## Mapping of disease-associated mutations to Pol I subunit structures rationalizes enzyme deficiencies

Four disease phenotypes were linked to mutation of Pol I subunits
in humans: acrofacial dysostosis (Cincinnati type) (69, 70), Treacher
Collins syndrome (TCS) (71, 72, 73, 74), hypomyelinating leukodys-
trophy (HL) (73, 75), and a juvenile neurodegenerative phenotype
akin to the HL-phenotype (76). With the structural model of hPol I
determined (Fig 2), we mapped these known mutations to gain
insight into the underlying molecular pathologies (Fig S5).

Acrofacial dysostosis, Cincinnati type, leads to craniofacial ab-
normalities during development and is caused by mutations E593Q
and V1299F in subunit RPA1 (69, 70). Mutation E593Q is located in
proximity to the catalytic center (Fig S5) and may directly affect the
nucleotide addition as indicated by a reported transcription sup-
pression of this mutant due to its enhanced rDNA binding stability
(70). In contrast, V1299F is situated on the interface of RPA1 with
RPA12 and may destabilize the association of this subunit with the
hPol I core (Fig S5).

Treacher Collins syndrome (TCS) is caused by various mutations
in the genes *TCOF1*, *POLR1B*, *POLR1C*, or *POLR1D*. S682 and R1003 of
RPA2 are likely to impair transcription by partially hindering
translocation or intrinsically destabilizing the subunit, respectively
(Fig S5). Other TCS-associated mutations within subunit RPAC2
(E47K, T50I, L51R, G52E, L55V, R56C, L82S, G99S) cluster at intra-
subunit and RPAC1 inter-subunit contacts (71, 72, 73) (Fig S5),
thus potentially destabilizing subunit assembly or leading to minor
defects in core stability. We conclude that polymerase-associated
TCS mutations can be functionally classified according to their
effects: (1) Impaired Pol I transcription activity (RPA2 mutations)
and (2) Effect on Pol I and Pol III transcription.

Similar to TCS, hypomyelinating leukodystrophy (HL) is a neu-
rodegenerative disease that cannot be classified as a Pol I- or Pol
III- associated disease per se. HL mutations are found in the shared
subunit RPAC1 and in Pol III specific subunits RPC1 and RPC2 (73, 75).
HL-associated mutations of RPAC1 (T26I, T27A, P30S, N32I, N74S, I105F,
H108Y, and R109H) apparently have a stronger effect on Pol III as they
are found in regions of shared subunit which are flexible in Pol I or are
part of Pol III-specific interaction surfaces (Fig S5). This is in line with
the finding that mutations N74S and N32I only affect Pol III assembly
but apparently do not impair Pol I biogenesis or nuclear import (73).
Additional HL-associated RPAC1 mutations (M65V, V94A, A117P, G132D,

C146R, R191Q, I262T, T313M, and E324K) are likely to affect RPAC1 folding itself and may therefore impact both, Pol III and Pol I (Fig S5).

Finally, the mutation S934L in RPA1 is associated with a juvenile neurodegenerative phenotype akin to the HL-phenotype associated with Pol III disruption (76). This mutation occurs in a small loop of RPA1 which forms contacts with RPA2 in the vicinity of the bridge helix and may destabilize contacts with RPA2 (Fig S5).

### A single-subunit stalk is the predominant configuration for Pol I

One of the major differences between Pol I enzymes of different organisms lies within the stalk subcomplex. DNA- and protein-sequence–based searches identified homologues for 13 of the 14 yeast Pol I subunits except for the stalk-subunit A14 (12). Divergence of the stalk subunits among DNA-dependent RNA polymerases is well documented. Compared with the Pol II stalk, a domain-swap between yeast Rpb4 and Rpb7 and the yeast Pol I stalk subunits A14 and A43 was observed in the crystal structure of the Pol I subcomplex (46, 77). With this swap, subunit A14 appears to harbor limited functional importance. Deletion of the subunit in *S. cerevisiae* is not lethal but results in conditional growth defects indicating regulation deficiencies (78, 79), similar to observations in *S. pombe* (80).

To analyze whether hPol I indeed carries a single-subunit stalk, mass spectrometric analysis of all protein bands in our purification was performed. The 13 subunits identified in situ and initiation factor RRN3 were found to be present with sequence coverages over 25% (Fig S2). Additional proteins were not identified with similar confidence. To clarify whether the absence of a second Pol I stalk subunit is specific to human cells and to understand the changed composition of the enzyme during its evolution, we carried out bioinformatic analysis: First, we generated a phylogenetic tree based on sequence similarity of the Pol I subunits RPA1, RPA34, and RPA43 to cover the polymerase core and the peripheral subcomplexes (Fig 3). Generating a Pol I-specific conservation tree removed bias that may originate from the influence of unrelated genes on global alignments in standard phylogenetic analysis. We clearly find that only organisms of the *Saccharomycotina* in the *Dikarya* clade carry sequences for the subunit A14, indicating that a single-subunit stalk is the pre-dominant Pol I configuration.

### Built-in transcription factors differ among organisms

Phylogenetic analysis also showed that the "expander" (DNA-mimicking) element is present in all analyzed organisms. This mobile insertion in the jaw domain of the largest subunit mimics DNA binding to inactive Pol I dimers (13, 14) or monomers (68).

The RPA49/34 heterodimer resembles the yeast A49/A34.5 subcomplex with functions in initiation and elongation (28, 44, 81) and is present in cryo-EM reconstructions. The subcomplex is related to the Pol II initiation factors TFIIF and TFIIE (15) and stays attached to the Pol I core throughout its transcription cycle in vivo (82), but may be lost under some conditions in vitro (46, 59, 68). The TFIIE-related, C-terminal tWH domain of subunit RPA49 is flexible in our reconstructions as expected for Pol I monomers and most elongation states. Similarly, we do not observe density for the mammalian-specific C-terminal extension of subunit RPA34 (compare Figs 2 and 3). This is also the case for a C-terminal extension of the hPol III

subunit RPC5 that contributes to enzyme stability despite being flexibly linked (20).

The C-terminal domain of RPA34 is enlarged to 55 kD in humans compared with the 27 kD yeast protein (Fig 2C and Supplemental Data 1). The C-terminal extension is present in higher organism classes, such as *Mammalia* and *Amphibia* but shows no clear conservation in sequence, predicted secondary structure or length (Fig 3), and is flexible in our cryo-EM reconstruction. To determine functional similarity with the yeast counterparts, we tested binding of recombinant human RPA49/34 to the *S. cerevisiae* enzyme purified from an A49 deletion strain resulting in a 12-subunit Pol I (Pol IΔ). Direct cross-species binding of the RPA49/34 heterodimer to Sc Pol I in vitro was not possible, likely due to divergence of the charged tail region ("ARM") of RPA34 and its binding site on the "external" domain of the second largest subunit RPA2.

In contrast to direct interaction, functional cross-species complementation of recombinant yeast and human subcomplexes was possible (Fig S3C). Recombinant Sc A49/34.5 and Hs RPA49/34 both recovered the activity of hPol IΔ in elongation and cleavage. Hence, interaction interfaces apparently co-evolved, while subcomplex function was retained from yeast to human. Both, Sc and Hs RPA49/34 can bind to DNA independent of core Pol I (Fig S6). While the main interface with DNA apparently lies within the TFIIE-related tWH domain of RPA49, the flexible and divergent RPA34 tail is capable of independent DNA-interaction. Notably, the elongation and cleavage pattern indicated no major differences depending on the type of heterodimer added (Sc or Hs version). Therefore, reduced proof-reading of hPol I apparently is an intrinsic enzymatic feature of the core enzyme rather than effects introduced by divergent heterodimer subunits or their sub-stoichiometric co-purification.

### A previously undescribed domain is built into the largest subunit of human Pol I

The second major difference between yeast and human Pol I is an insertion in the "foot" domain of the largest subunit RPA1 (Fig 2C and Supplemental Data 1). The Pol II foot domain serves as transient interaction platform for the regulatory co–activator complex "mediator" (83) and is enlarged compared with yeast Pol I (13, 14). This may lead to a speculation about a comparable regulatory role of the foot insertion specifically required in humans but not in yeast. We found well-defined cryo-EM density on the downstream face (front) of hPol I subunit RPABC1 (Rpb5) that is closely connected to the foot insertion site. Domain prediction using the HHPRED package (84) indicated a clear homology to a High Mobility Group ("HMG") box domain with the closest fit to the structure of HMG box 5 of the hPol I transcription factor UBF (85). Hence, we constructed a homology model of the foot insertion and fitted the resulting model into the observed cryo-EM density. This allows an unambiguous placement of the domain without adjustment, indicating that the hPol I foot insertion indeed resembles a built-in HMG box (Fig 4).

### The HMG box-containing "dock II" domain may serve as interface for Topoisomerase 2a

Canonical HMG box domains can bind the minor groove of a DNA duplex in a sequence-specific or unspecific manner with a

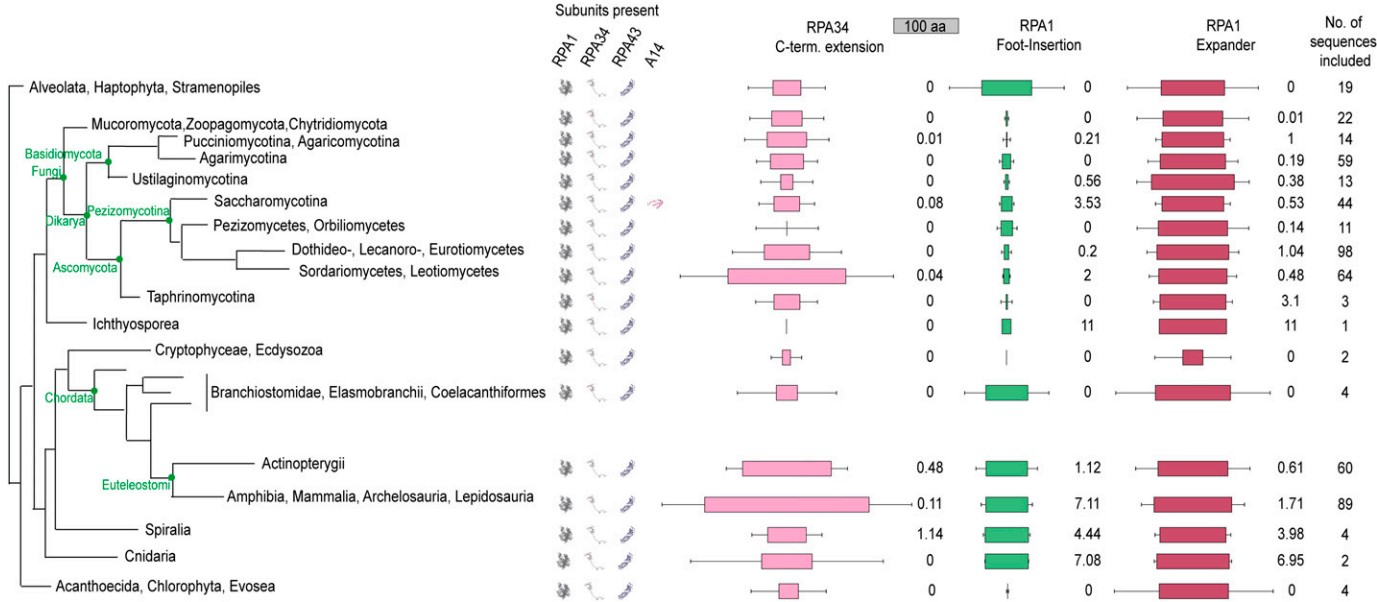

**Figure 3. Phylogenetic analysis of RNA polymerase I.**
Phylogenetic tree calculated based on sequence homology of the three Pol I subunits RPA1 (core), RPA34 (RPA49/34 heterodimer), and RPA43 (stalk subcomplex); schematic. The subunit A14 is found in all *Saccharomycotina* in the class of *Dikarya*. This includes model organisms such as *Saccharomyces cerevisiae* and *Schizosaccharomyces pombe*, explaining the current paradigm that Pol I comprises 14 subunits. Conservation scores for the RPA1 foot insertion, the Expander (DNA-mimicking loop), and the C-terminal extension of RPA34 were calculated in each class. Blocks show the median length of each specific region (100 residue referenced above). Box reflects the median; error bars indicate SD; conservation scores are grouped into five categories: not conserved (0–3), weakly conserved (3–5), medium conserved (5–7), conserved (7–9), and strongly conserved (9–11).

preference for non-B-form conformations (86). Overlay with a model HMG box (box 2 of the human HMGB1 protein) shows that the DNA-binding site of the hPol I foot insertion is completely occluded by the common Pol subunit RPABC1 (Fig 4C and D), indicating a divergent function. Furthermore, structure-based sequence alignment of the RPA1 foot HMG box shows that the so-called "minor wing" is absent. This minor wing consists of an N-terminal motif and the C-terminal extension of the HMG helix three (Fig 4E). Both regions cooperate in DNA-binding of canonical HMG boxes but are absent in RPA1. Furthermore, a loop between HMG box helices one and two directly interacts with DNA and contributes to sequence specificity (87). In RPA1, we observed an insertion between the corresponding helices α27d and α27e that contacts loop T56-V60 of subunit RPABC1 (Fig 4). In contrast, a basic surface patch is found on the opposite face (Fig S7). To test whether DNA-interaction is possible, we recombinantly expressed MBP-tagged versions of the domain (full length and minimal) and tested their ability to bind an unspecific dsDNA fragment. No major DNA-binding was observed for the minimal construct, whereas the full-length fragment showed residual DNA binding at 50× protein access in electron-mobility shift assays, likely due to an unspecific basic surface patch (Fig S7C). Therefore, we conclude that the RPA1 foot insertion represents a truncated HMG box "major wing" unable to bind DNA.

Apart from binding DNA, HMG boxes are known to promote protein–protein interactions. This could be a possible function of the RPA1 foot HMG box, which we hence termed "dock II." The human HMGB1 protein was found to interact with Topoisomerase (Top) 2a independent of DNA, whereas promoting the activity of this

enzyme (88). In fact, active Top2a co-purifies with the hPol I-RRN3 complex (89) and was described to be part of the hPol I transcription initiation machinery (90). Therefore, we asked whether recombinant human Top2a lacking the unstructured C-terminal domain (91) can interact with the RPA1 dock II domain. Indeed, we observe a shift in native PAGE of full length, but not minimal dock II or the MBP-tag alone, indicating the possibility for transient interaction (Fig S7E). To further validate the results of the native PAGE, we analyzed the bands by mass spectrometry. In the Top2a band of the control sample (incubated with MBP-tag only), indeed only Top2a, but not MBP, was found. Incubation with full-length MPB-dock II led to detection of the included RPA1- and MBP-peptides in both, the Top2a band and especially in the shifted band (Fig S7F).

**Molecular modelling identifies dock II as possible interaction site of Top2a**

To test whether binding of Top2a to hPol I using the dock II domain as interaction platform is theoretically possible, we carried out in situ molecular docking. hPol I fragments were docked to Top2a in its structurally resolved states I and II using HADDOCK (92), AutoDock Vina (93), the ZDOCK webserver (94), and PRISM webserver (95, 96) (see the Materials and Methods section). Interaction patterns were analyzed using the MAXIMOBY (CHEOPS) contact matrix algorithm and the VMD plugin PyContact (97) (for details, see the Materials and Methods section). Fig S8A shows the four most reliable results docking the complete RPA1 subunit to Top2a using the HADDOCK software. These results demonstrate that dock II-interaction with either the catalytic domain or the ATPase domain is possible (key

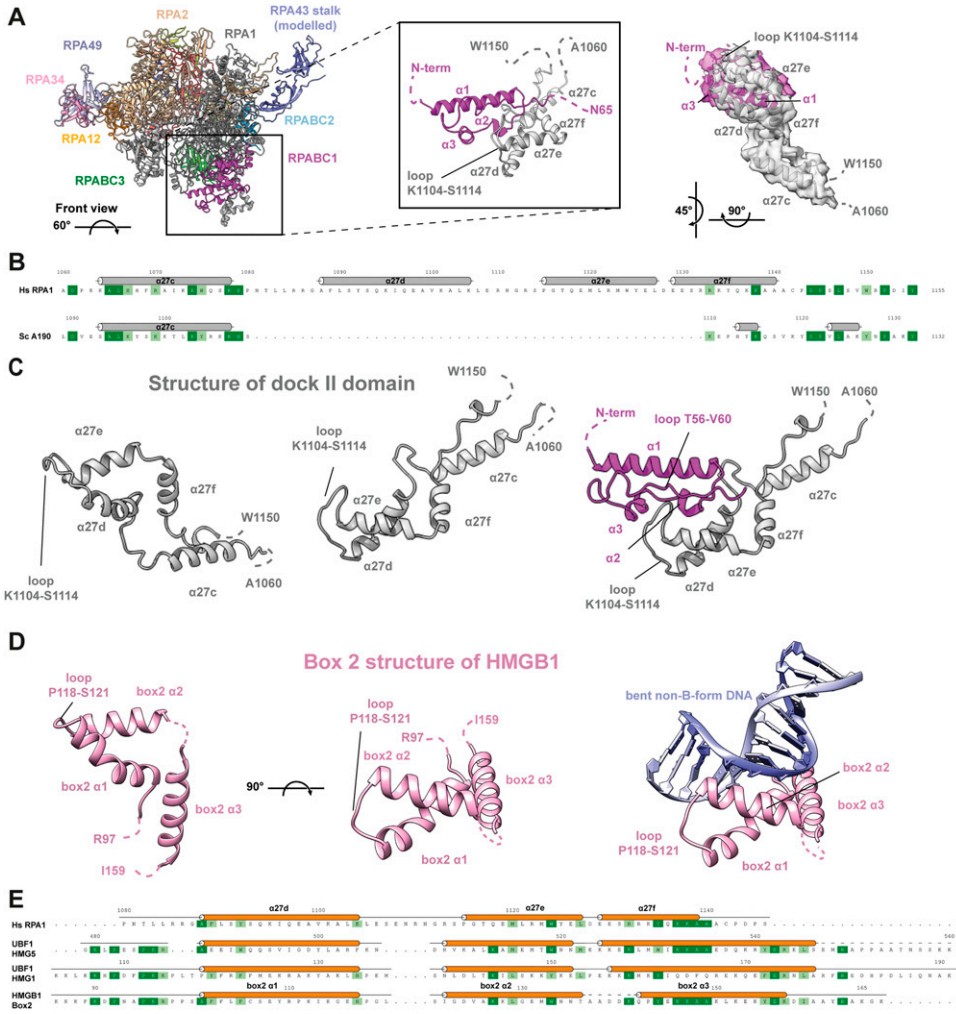

**Figure 4. An HMG box like domain is included into the largest subunit of human Pol I.**
**(A)** Location of the structured insertion in RPA1 α27d-f on the downstream edge of subunit RPABC1 in human Pol I and enlarged view of the region. Overlaid experimental cryo-EM density for the helices α27c-f of subunit RPA1 (grey) and the N-terminal 65 residues of RPABC1 (purple) shown as transparent surface (right). **(B)** Structure-based sequence alignment of human and yeast Pol I foot insertions (for complete sequence, compare Supplemental Data 1). **(C, D)** Structure of the RPA1-foot insertion (C) compared with the canonical HMG box 2 of the human protein HMGB1 (D) from two views. The DNA-binding surface of the canonical HMG box 2 is occluded by RPABC1 in hPol I. **(E)** Structure-based sequence alignment of the RPA1-HMG insertion with the canonical HMG box 2 of HMGB1, and the boxes 1 and 5 of the Pol I transcription factor UBF. In the RPA1-HMG box, the N-terminal region is divergent and the third helix is truncated. Both of these parts are important for DNA-interaction. A loop insertion between the first two helices is part of the RPABC1 interface.

residues of hypothetical docking sites listed in Table S2). A summary of all in situ docking results using the dock II domain only are shown in Fig S8B and are generally in line with HADDOCK results of complete RPA1. Interestingly, for all predicted structures that have no clashes between Top2a and hPol I the hPol I downstream DNA path is freely accessible.

Having established that a direct interaction of Top2a with hPol I via the dock II domain is possible, we asked whether Top2a would be present at the rDNA gene in cells and whether its distribution would resemble that of an initiation factor behavior as proposed previously (90). To this end, we re-analyzed previously published Top2a ChIP-Seq data from mouse cells (98) and mapped the initiation factor TAF1B (part of SL1 and homologous to TFIIB (99, 100)), UBF, Pol I (41), and Top2a to the rDNA gene as described (101). As shown in Fig 5A, TAF1B maps to clear peaks at the spacer promoter and the main rDNA promoter, defining the transcription start site (TSS). Pol I is distributed over the gene body and the spacer promoter, as expected in growing cells. Strikingly, Top2a maps to the rDNA locus but does not show the profile of a classical initiation factor, such as RRN3 which peaks at the promoter and tails out in the 5' region of the rDNA gene (41). Instead, Top2a is present over

the entire gene, with some peaks in the 3' region. These peaks apparently overlay with the UBF-binding sites.

## Physical interaction with UBF indicates functional cooperativity of Top2a and HMG box–containing proteins

Results from ChIP-Seq reanalysis do not exclude the possibility that Top2a is also part of some initiation complexes, but indicate either a Pol I-independent rDNA gene association, an elongation factor like behavior in cooperation with Pol I, and/or DNA-binding cooperativity with UBF. To test whether a physical interaction between UBF and Top2a takes place as indicated by co-localization of ChIP peaks, we performed immunoprecipitation assays from cell lysates using anti Top2a antibodies. These pull-downs confirmed a direct interaction between Top2a and hPol I demonstrated by Western blot against subunit RPA49. Furthermore, the observed signals for UBF are in line with an interaction in cells (Fig 5B).

To clarify whether UBF–Top2a interaction is direct, we tested the binding of recombinant FLAG-tagged UBF (fUBF) and Top2a. Incubation of both proteins in vitro followed by a pull-down using anti-FLAG antibodies showed a clear band for Top2a in Western blots

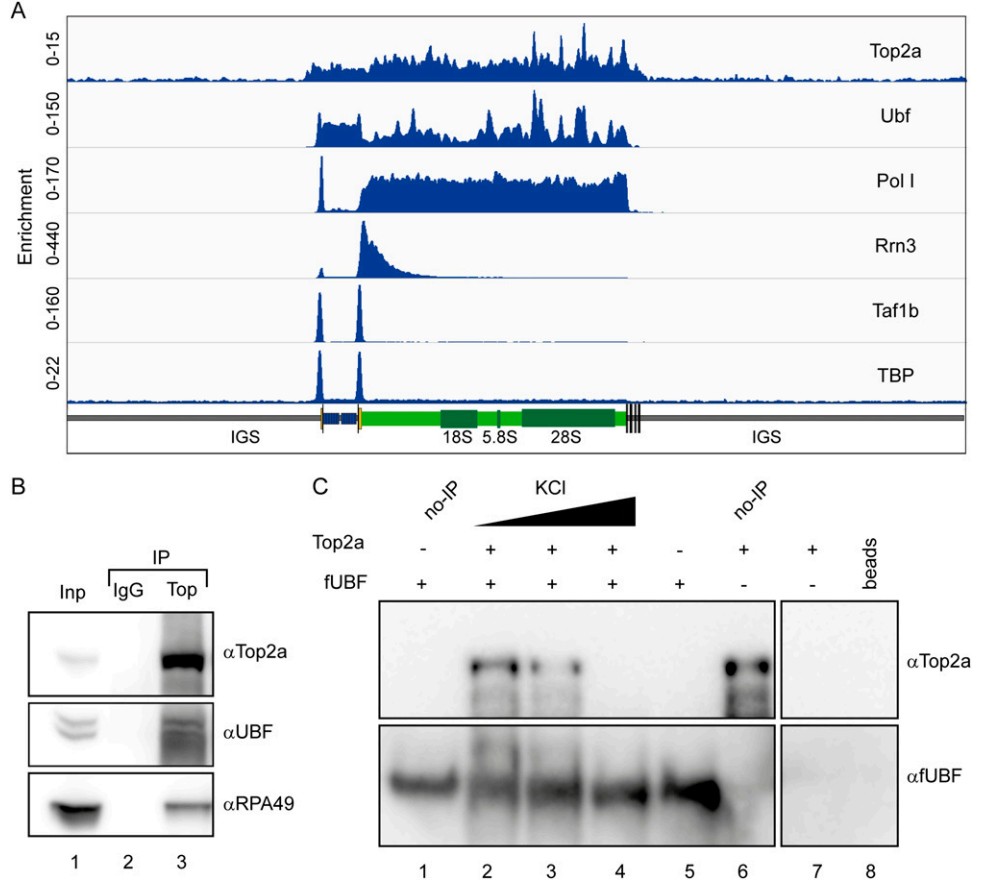

**Figure 5. Top2a localizes to the rDNA gene and interacts with UBF.**
**(A)** Top2a is detected over the entire mouse rDNA gene regions occupied by UBF. Original raw data from reference 98 was aligned and deconvoluted as previously described (101). Peaks over the 3′ region of the gene overlap with UBF peaks, indicating co-localization. Top2a overlaps binding peaks for the initiation factors RRN3, TAF1B and TBP, but specific correlations are not observed. Pol I signal marks the actively transcribed region. **(B)** UBF co-immunoprecipitates with Top2a: Top2a was immunoprecipitated from nuclear extract of U2OS cells using anti-Top2a antibodies (Abcam) immobilized on magnetic beads (DynaI). Immunoprecipitated proteins were analyzed by Western blot using anti-UBF, anti-RPA49, and anti-Top2a antibodies; Lane 1: 10% input, Lane 2: IP with IgG control, Lane 3: IP with anti-Top2a antibodies. **(C)** Purified Top2a co-precipitates with purified UBF at low salt concentrations. Recombinant fUBF was incubated with purified Top2a at three different salt concentrations (lanes 2–4). Lane 1: fUBF control (no IP), Lane 5: IP without Top2a, Lane 6: Top2a control (no IP), Lane 7: IP without fUBF addition, Lane 8: FLAG-bead only control.

(Fig 5C, lane 2). Increasing salt concentration weakened (lane 3, 100 mM KCl) and finally abolished (lane 4, 200 mM KCl) the co-IP. Notably, in situ docking studies using HMG box 5 of UBF identified similar binding sites as for dock II (Fig S8C). We conclude that Top2a can interact with both Pol I and UBF in human cells and in vitro.

## Discussion

The cryo-EM reconstruction of human Pol I demonstrates the overall conserved architecture of multi-subunit, DNA-dependent RNA polymerases in eukaryotes and completes the archive of yeast (13, 14, 16, 102) and mammalian (19, 20, 21, 22) nuclear Pol structures (Fig S9). We find that human Pol I, like that of most organisms, carries a single-subunit stalk, and built-in transcription factors show structural and functional similarities to TFIIF, TFIIE, and TFIIS. Mapping of known hPol I mutations associated with human disease to the structural model (Fig S5) rationalizes their effects on the enzyme. During initial peer-review of this work, two groups also reported cryo-EM reconstructions of hPol I (17, 18). The focus of one study lies on the structural basis of elongation and cleavage (17), whereas the other also reports a co-structure with RRN3 (18). Our colleagues present reconstructions with higher overall resolution, but do not comment on the role of the novel dock II domain and

involvement of Top2a in rDNA transcription. Therefore, the findings of the three studies support and supplement each other. We also show that functional cross-species complementation of RPA49/34 subcomplexes is possible, which is in line with a conserved role in supporting initiation and elongation stages of the transcription cycle while accumulating divergent regulatory properties (103, 104, 105). An increased flexibility of the clamp/stalk module in hPol I is indicated by the cryo-EM reconstruction (Fig 2) and may explain an increased rate of incorrect nucleotide addition we observe in comparison to the yeast enzymes in vitro (Fig 2B). This can be explained either by an impaired proof-reading due to reduced backtracking ability of hPol I, or a generally higher rate of substrate promiscuity. In yeast Pol I, module contraction is a feature of activation (106). Especially during DNA melting upon transcription initiation (107, 108), contraction is required to stably associate melted template and non-template strands. Notably, the catalytic center, including the active site magnesium ion, is among the flexible parts in the apo hPol I cryo-EM reconstruction, which is consistent with an already weak density for the open complex (18). The pronounced shelf module flexibility may indicate the importance of such a mechanism in higher eukaryotes, or simply point to a lack of defined intermediate conformations under close-to-native conditions in human cells.

Although we do not observe any cryo-EM density for bound human RRN3, high sequence conservation of the factor (24)

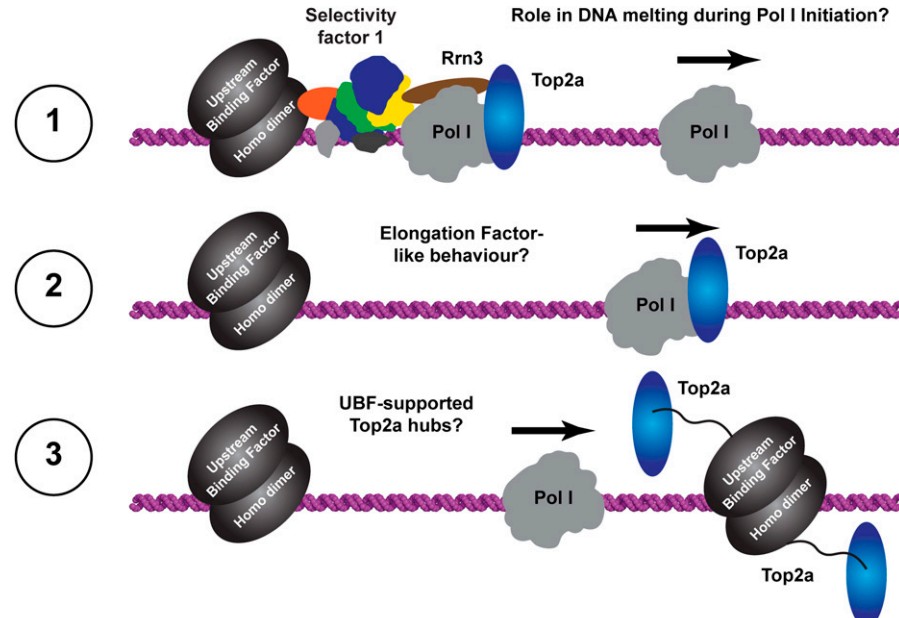

**Figure 6. Possible roles of Top2a in human Pol I transcription.**
Three hypotheses are likely scenarios: **(1)** Top2a may support initiation by resolving supercoils generated during dsDNA melting. **(2)** Top2a may travel with Pol I in an elongation factor like manner to resolve positive supercoils upon their accumulation. **(3)** Supported by direct and indirect evidence, we speculate that UBF and Top2a cooperate to form "torsion release hubs" at the 3′ region of the rDNA gene.

suggests that binding to hPol I subunit RPA43 and the dock domain of subunit RPA1 is similar to the *S. cerevisiae* counterpart (27, 28, 29). Yeast Pol I subunit A14 is not involved in Rrn3 contacts (27, 28, 29). Therefore, its absence in the human enzyme does not disagree with this model and the now available RRN3-bound reconstruction of hPol I confirms the overall conservation of binding modes while revealing a specific stalk-movement in hPol I (18). Notably, purification by ion exchange chromatography leads to a dissociation of human RRN3 and the RPA49/34 heterodimer (Fig S1B), indicating a reduced affinity and hence the possibility for efficient regulation of interaction with the core enzyme by post-translational modifications, such as RRN3 phosphorylation (109) and RPA49 acetylation (35).

Most strikingly, our study identifies a previously unknown built-in transcription factor–like domain that resembles the fold of a truncated HMG box (Fig 4). This "dock II" domain is only found in higher organisms (Fig 3) and shows similarities to HMG box 5 in UBF. Although its function will be studied in more detail in the future, we find evidence that it may serve as an interaction platform for human Topoisomerase 2a. Three possible reasons for this interaction come to mind (Fig 6): (1) Top2a could be part of Pol I initiation complexes in human cells (90), whereas it does not appear to be involved in yeast PIC formation. Top2a recruitment to the downstream edge of human Pol I PICs via the dock II domain and initiation factor RRN3 may be an attractive way to release tension from the DNA that accumulates upon spontaneous melting. In Pol II initiation systems, the XPB translocase in TFIIH occupies a similar position and carries out a comparable though not identical function in yeast (110) and human PICs (111, 112). Deletion of the Top2a C terminus leads to a sixfold reduction in RRN3 co-purification, but only a twofold reduction in hPol I co-purification (90), arguing for the possibility of a co-dependent Top2a recruitment via the foot-HMG box domain and RRN3. (2) Positive supercoiling accumulates in the direction of transcription (113), especially in Pol

I-transcribed rDNA genes (114), due to an increased loading rate (115) and speed compared with other polymerases (48). To release this supercoiling, Top2a may be recruited to the downstream face of elongating hPol I via the built-in HMG box. This may be reflected in an elongation factor–like behavior of Top2a and could be exclusive to the first round of transcription of a previously inactive rDNA gene. After Top2a-supported opening of the gene by initial hPol I transcription, including nucleosome removal assisted by FACT (116), association of UBF over the gene body (41) may prevent closing of active rDNA repeats and thus strong accumulation of positive supercoiling during subsequent rounds of Pol I transcription. (3) Association of UBF with Top2a over the rDNA gene may create periodic hubs that allow the transient recruitment and handover of Top2a between UBF and hPol I on active genes to release positive supercoiling. As indicated by domain similarity and in situ modelling, the three C-terminal HMG boxes of UBF may be responsible for such a Top2a interaction. In addition, UBF association with DNA introduces additional supercoiling itself (117). In actively transcribed genes, high on/off rates of UBF can be expected, leading to the local requirement of Top2a that could be satisfied by UBF association of the enzyme.

Options (2) and (3) are supported by the fact that Top2a signal is detected on the entire gene and co-localization of Top2a with UBF in some regions is observed in ChIP-Seq studies (Fig 5A). An initiation factor–like profile for Top2a that would point towards option (1) is not detected, which, however, does not exclude Top2a option (1). Though possibly coincidental, further evidence for hypothesis (3) arises from phylogenetic analysis demonstrating that UBF versions start to appear in the same organism in which we detect the presence of the dock II domain (Fig S10) and from the recent finding that Top2 localization to the nucleolus depends on Pol I activity in human cells (118). In line with this, we demonstrate that physical interaction between UBF and Top2a is possible.

Nevertheless, additional functions for the HMG box–containing dock II domain independent of Top2a can be imagined. The domain clashes with the "trestle" helix of the CTR9 subunit in the PAF-complex, a Pol II elongation factor (119). This may prevent PAF action in human Pol I transcription, even though an effect in yeast Pol I elongation was reported (120). Furthermore, the HMG box–containing SOX factors assist DNA detachment from nucleosomes (121), and SSRP1 is a component of FACT that also contains a single HMG box and is required for hPol I transcription through nucleosomes (116). In fact, single HMG box-containing proteins were described to functionally support human FACT (122). Together with the positioning of dock II close to the incoming (downstream) DNA duplex, this also supports the speculation of a function in efficient nucleosome encounter of hPol I. Most of these factors, however, require a direct DNA interaction of their HMG box, which appears unlikely for dock II because of occlusion of the DNA interface by RPABC1 and its mutated DNA-binding site (Fig 4).

Although this work suggests previously undescribed structural and functional links between Top2a and Pol I action in human cells, the mechanistic basis for the interaction still needs to be deciphered and many questions remain. Why would increased Top2a activity be necessary, especially at Pol I genes? Is Top2a recruitment dependent on UBF and the dock II interface of Pol I? Functional correlation of Top2a occupancy and mutational studies in yeast and human transcription systems will provide the answers in the future. However, it is not too surprising to find another transcription factor–related domain built into metazoan Pol I. In addition to TFIIF and TFIIE elements within the Pol I-specific subunits RPA49/34, TFIIS elements in subunit RPA12 and a DNA-mimicking element in RPA1, the integration of an HMG box element seems to contribute to the accumulating specialization. Even though none of these adaptations are essential, their sum defines the adaptation of the transcription system to its unique task throughout evolution.

# Materials and Methods

### CRISPR/Cas9 genome editing

HeLa cells were cultivated in DMEM medium (21885; Gibco) supplemented with 10% FBS (10270; Gibco) and 1% Penicillin/Streptomycin (P0781; Sigma-Aldrich) at 37°C and 5% $CO_2$ atmosphere. Genomic integration of sfGFP ORF at the C terminus of RPA1 was done by CRISPR/Cas9 according to a published protocol (123) with some modifications and identical as previously published for RPAC1-sfGFP (20).

Design of the gRNAs was done with a Web-based tool (https://www.benchling.com/crispr/) and annealed oligonucleotides (gRNA1 = GCTCCAAGGACCCTTGGTGA; gRNA2 = CGGGGTAGCTGCTATCTCAG) were cloned via BbsI as described in the manual into the Cas9n expression vector pSpCas9n(BB)-2A-Puro (PX462) V2.0, which was a gift from Feng Zhang (plasmid #62987; Addgene; https://www.addgene.org/62987/; RRID: Addgene_62987). A donor plasmid carried a short GS-linker sequence with an embedded HRV 3C protease cleavage site and the sfGFP ORF surrounded by two large sequence segments homologous to the insertion locus in the genome.

HeLa cells were transfected with a 1:1:1 M ratio of gRNA1 and gRNA2 vectors together with the donor plasmid using FuGENE HD Transfection Reagent (E2311; Promega) according to the manufacturer's instructions. Several days later, the GFP-expressing cells were enriched by flow cytometry using a BD FACSAria IIu cell sorter at the Central FACS Facility of the RCI Regensburg (Center for Interventional Immunology). GFP-positive cells were seeded as single cells on 96-well plates. After 2–3 wk, colonies were expanded. These monoclonal populations were validated for the tag insertion by PCR on extracted genomic DNA (gDNA), sequencing and Western blot.

About $1 \times 10^6$ cell were resuspended in proteinase K buffer (20 mM Tris, pH 7.5, 300 mM NaCl, 25 mM EDTA, 2% [wt/vol] SDS, and 0.2 mg/ml proteinase K) and incubated overnight at 50°C before performing isopropanol precipitation. The resuspended gDNA was used as template for PCR to validate the homozygous introduction of the GS-linker and sfGFP ORF into the POLR1A genomic locus (Primer: POLR1A-fwd1: 5'-TTGGGATCCGGTCAAACTC-3', POLR1A-rev1: 5'-#CAGCAAAGCATGGCTTCC-3', POLR1A-fwd2: 5'-CAGTGGGATCTTGG-GATCTG-3', POLR1A-rev2: 5'-TGCTACGCTGTACTTGACTC-3'). To further validate the result, the PCR product was gel extracted (QIAquick Gel Extraction Kit, 28706; QIAGEN) and sequenced (Microsynth Seqlab). Additional characterization of the selected homozygous cell line was performed by Western blot. Cells from a confluent 6-cm plate (about $2.7 \times 10^6$ cells, 83.3901.300; Sarstedt) were harvested with 300 $\mu$l of boiling 1× SDS loading dye (3% [wt/vol] glycerol, 1.68% [vol/vol] $\beta$-mercaptoethanol, 0.03% [wt/vol] bromophenol blue, 26 mM Tris, pH 6.8, and 0.42% [wt/vol] SDS) and vigorously shaken at 95°C for 15 min. Prestained marker (7719S; NEB), as well as 10 $\mu$l of sample from the parental and the newly generated cell line, were loaded on an SDS gel (NP0223BOX; Thermo Fisher Scientific) and proteins were separated by electrophoresis. After blotting (Trans-Turbo Blot; Bio-Rad) the proteins onto a PVDF membrane (1704275; Bio-Rad), Ponceau S staining confirmed equal loading. The tagged protein RPA1 was detected by the primary antibody (sc-48385; Santa Cruz Biotechnology), which was subsequently detected by the fluorescently labeled secondary antibody (926-32210; Li-COR). Prestained marker and secondary antibody were detected by different wavelengths (Odyssey Infrared Imager Model 9120; Li-COR).

The selected cell line POLR1A-sfGFP was cultivated adherently and adapted to suspension growth as follows: Cells from eight flasks (about $7 \times 10^7$ cells total; 83.3912.302; Sarstedt) were detached by incubation with trypsin (25300; Gibco) at 37°C for 5 min, transferred to a spinner flask (250 ml total volume; 4500; Corning), and cultured in suspension with high-glucose DMEM (11965; Gibco) supplemented with 1% FBS (10270; Gibco) and 1% penicillin/streptomycin (P0781; Sigma-Aldrich) under moderate stirring at 37°C and 5% $CO_2$ atmosphere. To expand the culture, 1× the current volume of fresh media including all supplements was added when the cells reached a density of ~$7 \times 10^5$ cells/ml and the culture was transferred to spinner flasks of increasing volume when required. Cells were harvested by centrifugation and washed with PBS before flash-freezing the pellet.

### Purification of human Pol I

Human Pol I purification was performed similarly to reference 20 with some modifications. POLR1A-sfGFP cell pellet was resuspended in twice the volume of the cell pellet's weight of lysis buffer

(20 mM Hepes, pH 7.8, 420 mM NaCl, 1 mM MgCl$_2$, 10 $\mu$M ZnCl$_2$, 0.5% [vol/vol] NP-40, 4 mM $\beta$-mercaptoethanol, and 1× protease inhibitor mix [Benzamidine & PMSF]) supplemented with 7 U/ml DNase I (M610A; Promega) and lysed by Dounce homogenization and incubation on ice for 30 min. After centrifugation at 20,000$g$ and 4°C for 15 min, the whole-cell lysate was incubated with pre-equilibrated GFP-Trap Dynabeads (gtd; ChromoTek) for binding. The beads were washed once with four times and once with twice the slurry volume of wash buffer (20 mM Hepes, pH 7.8, 420 mM NaCl, 1 mM MgCl$_2$, 10 $\mu$M ZnCl$_2$, 2% [vol/vol] glycerol, and 4 mM $\beta$-mercaptoethanol), before being eluted with the volume of the slurry with wash buffer supplemented with 10 $\mu$g of 3C protease per 1 g of cell pellet for 4 h at 4°C. In case an anion-exchange chromatography was performed, the GFP elution was diluted with buffer A (20 mM Hepes, pH 7.8, 1 mM MgCl$_2$, 10 $\mu$M ZnCl$_2$, 2% [vol/vol] glycerol, and 5 mM DTT) to reach a final concentration of 140 mM NaCl. The sample was loaded on a MonoQ 1.6/5 PC column (Pharmacia Biotech) with 60 mM ammonium sulfate and eluted stepwise in buffer A with increasing the concentration of ammonium sulfate up to 1 M. A linear gradient over five column volumes to 200 mM followed by steps of five column volumes with 200, 350, 600 mM and 1 M ammonium sulfate was applied. hPol I eluted at 350 mM ammonium sulfate concentration. hPol I was used immediately or flash-frozen in liquid nitrogen and stored at −80°C for further experiments.

### RNA elongation and cleavage assay

RNA Elongation and Cleavage Assay was performed as described (20) with small modifications. 0.5 pmol of Pol I from *S. cerevisiae*, *S. pombe*, or *Homo sapiens* were preincubated with 0.25 pmol of different pre-annealed minimal or bubble nucleic acid scaffolds (sequence information summarized in Table S3 and schematically shown in each figure along with the gel) in transcription buffer (20 mM Hepes, pH 7.8, 40 mM (NH$_4$)$_2$SO$_4$, 28 mM NaCl, 8 mM MgSO$_4$, 10 $\mu$M ZnCl$_2$, 10% [vol/vol] glycerol, and 10 mM DTT) for 1 h at 20°C in a 45 $\mu$l reaction. In case purified RPA49/RPA34 heterodimer was added, 1×, 5×, or 10× molar excess of heterodimer compared with polymerase was included during the preincubation. For RNA elongation, 10 $\mu$mol of each desired NTP (marked specifically at each lane in the figure) were added and the reaction was incubated for 1 h at 28°C. To examine cleavage activity, the preincubated reaction was incubated for 1 h at 28°C without the addition of NTPs. Afterwards, nucleic acid purification was examined by adding 5M NaCl to a final concentration of 0.5 M and 800 $\mu$l 100% ethanol. After precipitation for at least 1 h at −20°C, the sample was centrifuged for 30 min at 20,000$g$ and 4°C. The pellet was washed with 80% ethanol and, after drying, resuspended in 1× RNA loading dye (4 M urea, 1× TBE, 0.01% bromophenol blue, and 0.01% xylene cyanol only for FAM-labeled constructs). The sample was heated to 95°C for 5 min. As control 0.25 pmol of scaffold were treated identically, without addition of polymerase and NTPs. 0.125 pmol of FAM-labeled RNA product were separated by gel electrophoresis (20% polyacrylamide gel containing 7 M urea) and visualized with a Typhoon FLA9500 (GE Healthcare).

### Purification of RPA49/RPA34 variants

The *S. cerevisiae* full-length heterodimer was purified as described (15). Sc A49 with a C-terminal hexa-histidine tag and Sc A34 were co-

expressed in *Escherichia coli* BL21 (DE3) RIL in LB medium with 0.2 mM IPTG for 18 h at 18°C. The cells were resuspended in lysis buffer (50 mM Tris, pH 7.5, 300 mM NaCl, 10 mM $\beta$-mercaptoethanol, and 1× protease inhibitor [PI] mix [Benzamidine & PMSF]) and sonified. After centrifugation, the lysate was loaded onto preequilibrated Ni-NTA beads (30230; QIAGEN) by gravity-flow, washed with six times the bed volume of buffer Wash I (50 mM Tris pH 7.5, 1 M NaCl, 10 mM $\beta$-mercaptoethanol, 1× PI), and six times the bed volume of Wash II (50 mM Tris pH 7.5, 300 mM NaCl, 30 mM imidazole, 10 mM $\beta$-mercaptoethanol, and 1× PI) before elution (50 mM Tris, pH 7.5, 300 mM NaCl, 100 mM imidazole, 10 mM $\beta$-mercaptoethanol, and 1× PI). The sample was diluted threefold with dilution buffer (50 mM Tris, pH 7.5, and 10 mM $\beta$-mercaptoethanol) before loading onto a MonoS 5/50 GL column (GE Healthcare) with buffer A (50 mM Tris, pH 7.5, 100 mM NaCl, and 5 mM DTT). Elution was performed with a linear gradient of NaCl concentration up to 1 M. Sc A49/34 eluted at around 280 mM NaCl. The corresponding fractions were pooled and concentrated with 10 kD cut off (UFC801024; Millipore) and applied to a Superdex200 Increase 100/300 (GE Healthcare) equilibrated with buffer A. Pooled peak fractions were concentrated and flash-frozen for storage at −80°C.

The different variants of the human heterodimer (RPA49$^{FL}$/RPA34$^{FL}$, RPA49$^{FL}$/RPA34$^{1–343}$, RPA34$^{131–510}$) were cloned with an N-terminal 6xHis-tag on RPA49 and untagged RPA34, except for RPA34$^{131–510}$, which carries an N-terminal 6xHis-tag itself. The proteins were coexpressed in *E. coli* BL21 (DE3) RIL in LB medium with 0.2 mM IPTG overnight at 18°C. Cells were resuspended in lysis buffer (50 mM MES, pH 6.3, 300 mM NaCl, 10 mM $\beta$-mercaptoethanol, and 1× protease inhibitor [PI] mix [Benzamidine & PMSF]) and lysed by sonification. After centrifugation, the lysate was loaded onto preequilibrated Ni-NTA beads (30230; QIAGEN) by gravity-flow, washed subsequently with six times the bed volume of buffer Wash I (50 mM MES, pH 6.3, 1 M NaCl, 10 mM $\beta$-mercaptoethanol, and 1× PI), ATP-Wash (50 mM MES, pH 6.3, 1 M NaCl, 10 mM $\beta$-mercaptoethanol, 1× PI supplemented with 2 mg/ml denatured proteins, and 0.5 mM ATP), another ATP-Wash after 10 min of incubation and Wash II (50 mM MES, pH 6.3, 300 mM NaCl, 10 mM imidazole, 10 mM $\beta$-mercaptoethanol, and 1× PI) before elution (50 mM MES, pH 6.3, 300 mM NaCl, 200 mM imidazole, 10 mM $\beta$-mercaptoethanol, and 1× PI). The ATP-Wash steps were performed at room temperature. The sample was diluted fivefold with buffer A (50 mM Tris, pH 7.5, and 10 mM $\beta$-mercaptoethanol) before loading onto a MonoS 5/50 GL column (GE Healthcare) with buffer A supplemented with 100 mM NaCl. Elution was performed with a linear gradient of NaCl concentration up to 2 M. The corresponding fractions were pooled and concentrated with 10 kD cut off (UFC801024; Millipore) and applied to a Superdex200 Increase 100/300 (GE Healthcare) equilibrated with SEC buffer (50 mM Tris, pH 7.5, 150 mM NaCl, and 5 mM DTT). Pooled peak fractions were concentrated and flash-frozen for storage at −80°C.

### Purification of recombinant dock II domain

Two variants of the human dock II domain (RPA1$^{1060–1155}$ [full-length], RPA1$^{1081–1146}$ [minimal]) were cloned with a C-terminal His-MBP-tag. The proteins as well as tag-only were expressed overnight at 20°C in *E. coli* BL21 (DE3) RIL in LB medium with 0.2 mM IPTG. Cells were resuspended in lysis buffer (50 mM MES, pH 6.3, 300

mM NaCl, 10 mM β-mercaptoethanol, and 1× protease inhibitor [PI] mix [Benzamidine & PMSF]) and lysed by sonification. After centrifugation, the lysate was loaded onto preequilibrated Ni-NTA beads (30230; QIAGEN) by gravity-flow, washed subsequently with six times the bed volume of buffer Wash I (50 mM MES, pH 6.3, 1 M NaCl, 10 mM β-mercaptoethanol, and 1× PI) and Wash II (50 mM MES, pH 6.3, 300 mM NaCl, 10 mM imidazole, 10 mM β-mercaptoethanol, and 1× PI) before elution (50 mM MES, pH 6.3, 300 mM NaCl, 200 mM imidazole, 10 mM β-mercaptoethanol, and 1× PI). The eluent was buffer-exchanged to SEC buffer (50 mM Tris, pH 7.5, 150 mM NaCl, and 5 mM DTT) with a PD10 column (17-0850-01; GE Healthcare) and applied to a Superdex 75 Increase 10/300 GL (GE Healthcare) equilibrated with SEC buffer. Pooled peak fractions were concentrated and flash-frozen for storage at –80°C.

### Electrophoretic mobility shift assay

A total of 100 fmol pre-annealed 40 bp DNA (EMSA-DNA-strand1: 5′-Cy5- CTGGAACAACACTCAACCCTATCTCGGTCTATTCTTTTGA-3′; EMSA-DNA-strand2: 5′-TCAAAAGAATAGACCGAGATAGGGTTGAGTGTTGTTCC-AG-3′) were mixed with up to 50-fold molar excess of purified protein (as labeled in the figure) in EMSA buffer 1 or 2 (EMSA-buffer-1: 10 mM Tris, pH 7.5, 50 mM NaCl, 1 mM MgCl$_2$, 4% glycerol, 0.5 mM EDTA, 0.5 mM DTT; EMSA-buffer-2: 20 mM Hepes, pH 7.8, 150 mM NaCl, 2% glycerol, 0. 2% Triton-100, 0.2% Tween-20, and 5 mM DTT) and incubated at room temperature for 30 min. Afterwards 6× loading dye (10 mM Tris, pH 7.6, 60 mM EDTA, 60% glycerol, and 0.03% Orange G) was added to reach 1× concentration. 10% polyacrylamide gels in 0.4× TBE were pre-run at 110 V for 30 min before the reaction was separated at 110 V for 1:45 h at 4°C. The Cy5-labeled DNA was detected with a Typhoon FLA9500 (GE Healthcare).

### Confocal microscopy

For fluorescence imaging, cells were grown adherently on glass cover slips to 50% confluency. After washing the cells with pre-warmed (37°C) PBS, they were fixed with 3.7% paraformaldehyde in PBS for 10 min at 37°C. The fixation was stopped by replacing the solution with 100 mM glycine in PBS for 5 min at 37°C. After that, the cells were washed twice with PBS, mounted on the specimen slide with the help of a drop of Prolong Gold Antifade Mountant with DAPI (P36941; Thermo Fisher Scientific), and dried in the dark at least overnight.

The fluorescent specimens were imaged using a Plan-Apochromat 63×/1,4 Oil DIC Objective at a Zeiss LSM980/Airyscan 2 confocal microscope. sfGFP was excited by a 488-nm diode laser and emission was detected using a 300–720-nm band pass filter. Separately, DAPI was excited by a 405-nm diode laser and emission was detected using a 300–720-nm band pass filter. For the 3D model, a Z-stack was imaged using the internal GaAsP-PMT detectors from 490 to 668 nm for sfGFP and 410–473 nm for DAPI in a two-track process. Image processing was carried out using the Zeiss AxioVision software. The 3D Volume images were created in Imaris 9.6.

### Analysis of Pol I subunits RPA1, RPA34, RPA43, and A14

Data sets from Pol I subunits were generated using their corresponding InterPro (124) entries (RPA1: IPR015699, RPA34: IPR013240,

RPA43: IPR041901, and IPR041178, A14: IPR013239 downloaded on 07.06.2021). A common data set of RPA1, RPA34, and RPA43 was generated by searching for common species within the three InterPro families. To each obtained species, the concatenated sequence of RPA1, RPA34, and RPA43 was assigned.

### Phylogenetic analysis

Sequence alignment tool MAFFT (125) has been used with default options and a gap open penalty of 70. The resulting alignment was filtered manually on highly diverged sequences. To improve the quality of the phylogenetic analysis without losing information for each genus, only one sequence was chosen. On the resulting data set with 513 sequences, Gblocks (126) (options: b3 = 5,000, b4 = 2, b5 = a) has been applied to remove uninformative columns. By means of RAxML (127) using the option -f a and the substitution model PROTGAMMAAUTO, 100 trees were generated and a consensus tree was derived. The root has been placed between the supergroups of Sar and Haptophyta and the supergroup of Amorphea (128). The resulting phylogenetic tree was analyzed with respect to the taxonomic distribution. Sequences were grouped according to branching points in the phylogenetic tree (Fig 3). To retrieve the taxonomic group where the A14 subunit is present, the species related to the A14 subunit InterPro entry are compared with the species given in the phylogenetic tree.

### Sequence analysis of RPA34 and RPA1

By means of MAFFT sequence alignment of each subunit was generated using varied gap open penalties (RPA34: 50, RPA1: 20). Because of higher sequence variety within RPA34 sequences, BLOSUM30 was used instead of the default parameter. To account the divergence between the taxonomic groups given from the phylogenetic tree, the alignment was split into these groups and each group was analyzed separately on the presence or absence of the RPA34 C-terminal extension, the RPA1 foot domain and the RPA1 expander domain. Sequences from *H. sapiens* have been used as reference to identify the region of interests (399–510; 1,074–1,139; 1,365–1,488, respectively). The median length and SD of the regions of interest have been calculated for each group. To unravel the sequence and structural conservation of the regions of interest, the conservation score given in Jalview (129) has been extracted after removing all columns containing only gaps. The mean conservation score is calculated by summing up over all column scores divided by the number of columns. Scores are grouped into five categories: not conserved (0–3), weakly conserved (3–5), medium conserved (5–7), conserved (7–9), and strongly conserved (9–11). Secondary structures were predicted using Ali2D (84, 130). Secondary structure elements were assigned when more than five amino acids have medium to high probability in more than 90% of the sequences within each group. Bridging of two secondary structure elements over less than five differently annotated amino acids are counted as one element. If gaps are present in more than 90% of the sequences, they are ignored.

### Mass spectrometry

Protein bands were cut out from the gel, washed with 50 mM NH$_4$HCO$_3$, 50 mM NH$_4$HCO$_3$/acetonitrile (3/1), 50 mM NH$_4$HCO$_3$/

acetonitrile (1/1), and lyophilized. After a reduction/alkylation treatment and additional washing steps, proteins were *in gel* digested with trypsin (Trypsin Gold, mass spectrometry grade; Promega) overnight at 37°C. The resulting peptides were sequentially extracted with 50 mM $NH_4HCO_3$ and 50 mM $NH_4HCO_3$ in 50% acetonitrile. After lyophilization, peptides were reconstituted in 20 $\mu$l 1% TFA and separated by reversed-phase chromatography. An UltiMate 3000 RSLCnano System (Thermo Fisher Scientific) equipped with a C18 Acclaim Pepmap100 preconcentration column (100 $\mu$m i.D. ×20 mm; Thermo Fisher Scientific) and an Acclaim Pepmap100 C18 nano-column (75 $\mu$m i.d. ×250 mm; Thermo Fisher Scientific) was operated at a flow rate of 300 nl/min and a 60 min linear gradient of 4 to 40% acetonitrile in 0.1% formic acid. The LC was online-coupled to a maXis plus UHR-QTOF System (Bruker Daltonics) via a CaptiveSpray nanoflow electrospray source. Acquisition of MS/MS spectra after CID fragmentation was performed in data-dependent mode at a resolution of 60,000. The precursor scan rate was 2 Hz processing a mass range between m/z 175 and m/z 2,000. A dynamic method with a fixed cycle time of 3 s was applied via the Compass 1.7 acquisition and processing software (Bruker Daltonics). Before database searching with Protein Scape 3.1.3 (Bruker Daltonics) connected to Mascot 2.5.1 (Matrix Science), raw data were processed in Data Analysis 4.2 (Bruker Daltonics). Swiss-Prot *H. sapiens* database (release-2020_01, 220420 entries) was used for database search with the following parameters: enzyme specificity trypsin with one missed cleavage allowed, precursor tolerance 0.02 D, MS/MS tolerance 0.04 D, and Mascot peptide ion-score cut-off 25. Deamidation of asparagine and glutamine, oxidation of methionine, carbamidomethylation or propionamide modification of cysteine were set as variable modifications.

## Native PAGE

To investigate protein–protein interaction, blue-native PAGE was performed. Five times molar excess of MBP-only or tagged human dock II domain was incubated with recombinant Top2a ΔC (1–1,217) in binding buffer (20 mM Hepes, pH 8.0, 150 mM NaCl, 50 mM KCl, 1 mM MgCl2, 2% glycerol, and 2 mM $\beta$-mercaptoethanol) for 30 min at room temperature. After adding NativePAGE sample buffer, the samples were separated on a Native PAGE 3–12% gradient gel at 150 V for 90 min with light blue cathode and anode buffer (NativePAGE Novex Bis-Tris Gel System, BN1003BOX, Novex) and Coomassie stained.

## Top2a co-immunoprecipitation

To investigate Top2a interaction partners, co-immunoprecipitation was performed from U2OS Nuclear Extract (15 mg/ml total protein). Top2a was immunoprecipitated using an anti-Top2a antibody (ab12318; Abcam) immobilized on Dynabeads Protein A magnetic beads (c/n 10001D; Thermo Fisher Scientific) according to the manufacturer's instruction. Antibodies were cross-linked to beads using DPM (c/n 21666; Thermo Fisher Scientific) as recommended by the manufacturer. Beads were blocked with BSA in PBS overnight. 100 $\mu$l NE was diluted by dilution buffer (25 mM Tris HCl, pH 7.9, 12.5 mM $MgCl_2$, 10% glycerol, and 0.03% NP40) to a final KCl concentration of 150 mM and treated by 500 U of benzonase (E1014; Sigma-Aldrich) for 30 min at 4°C. 25 $\mu$l of the beads were added, and the suspension was incubated on a rotating wheel for 1 h at 4°C.

Beads were washed three times with 100 $\mu$l wash buffer (25 mM Tris–HCl, pH 7.9, 150 mM KCl, 12.5 mM $MgCl_2$, 10% glycerol, and 0.03% NP40) and proteins were eluted by incubation in 1× LDS sample buffer (c/n NP0007; Thermo Fisher Scientific) at 65°C for 10 min. Immunoprecipitated proteins were analyzed by Western blot using anti-UBF, anti-RPA49, and anti-Top2a antibodies (sc-9131; Santa Cruz; 611413 BD Transduction; and ab12318; Abcam).

## UBF-Top2a pull-down

To investigate protein–protein interaction, a pull-down assay using purified recombinant Flag-tagged UBF (fUBF) and purified Top2a was performed. fUBF was expressed in insect cells and purified as described earlier (131). Top2a was obtained from Inspiralis (c/n HT210). Proteins were incubated together in pull-down buffer (25 mM Tris–HCl, pH 7.9, 12.5 mM $MgCl_2$, 10% glycerol, and 0.03% NP40 supplemented with 50,100, or 200 mM KCl as marked in the Fig 5C) for 20 min at 4°C. To each sample, 20 $\mu$l anti-FLAG M2 Magnetic Beads (M8823; Sigma-Aldrich) were added and the suspension was incubated on a rotating wheel for 30 min at 4°C. Beads were washed three times with wash buffer (25 mM Tris–HCl, pH 7.9, 12.5 mM $MgCl_2$, 10% glycerol, and 0.03% NP40 supplemented with 50,100, or 200 mM KCl) and proteins were eluted by incubation in 1× LDS sample buffer (c/n NP0007; Thermo Fisher Scientific) at 65°C for 10 min. Proteins were analyzed by Western Blot using anti-UBF and anti-Top2a antibodies (sc-9131; Santa Cruz; ab12318; Abcam).

## Reanalysis of previously published ChIP data sets

Raw data were handled, mapping coordinates exacted, and the data displayed as previously published (101). The used data were as follows: Top2A GSE99197_SRR5585950_TOP2A-MEF (98). ArrayExpress E-MTAB-5839 data sets were as follows: ChIP-seq_UBF_MEFs_UBFfl_Rep1; ChIP-seq_RPI_MEFs_UBFfl_Rep1; ChIP-seq_Rrn3_MEFs_UBFfl_Rep1; ChIP-seq_TBP_MEFs_UBFfl_Rep1; and ChIP-seq_TAF68_MEFs_UBFfl_Rep1 (41). Taf1c is not included in the figure because it is identical to the Taf1b mapping, but data are also available in E-MTAB-5839 as ChIP-seq_TAF95_MEFs_UBFfl_Rep1.

## Negative stain EM

hPol I samples were centrifuged (4°C; 15,000 rpm; Eppendorf table top centrifuge) for 5 min. Five $\mu$l of the samples were then applied to glow-discharged 400-mesh copper grids (G2400C; Plano) with a self-made carbon film of ~7 nm thickness (60). After 1 min, the grids were washed in ddH$_2$O for 30 s, and stained three times with 5 $\mu$l saturated uranyl formiate solution (2 × 20 s, 1 × 30 s). After each step, excess liquid was removed with a filter paper. Images were collected on a JEOL 2100-F Transmission Electron Microscope operated at 200 keV and equipped with TVIPS-F416 (4k × 4k) CMOS-detector at 40,000× magnification (pixel size 2.7 Å) with alternating defocus (–1 to –3 $\mu$m).

The images were processed using RELION 3.1 (62) as shown in Fig S1. A total of 76 micrographs were analyzed, yielding 46,196 auto-picked particles using Laplacian-of-Gaussian (LoG) routine. After reference-free 2D sorting, a 3D classification (reference PDB: 5M3M low-pass filtered to 60 Å) yielded three reconstructions with different clamp/stalk flexibilities (Fig S1).

### Cryo-EM grid preparation and data collection

Reconstructions suffered from poor Fourier completeness. Screening for suitable conditions using crosslinking, gradient fixation (132) and detergents, or variation of grid support types graphene (-oxide), ultrathin carbon or gold foil (60) had limited success in removing orientational bias. Tilted data collection partially improved the bias even though 3D reconstruction was still hampered. Nevertheless, best results were obtained with GFP trap–eluted sample directly applied to graphene oxide–supported grids. However, this strategy retains some remaining 3C protease in the sample (Figs 1C and S1A) that may have a negative influence on signal-to-noise ratio.

Graphene oxide grids were prepared using the surface assembly method on Quantifoil R1.2/1.3 grids (133). Three microliters of sample were applied and incubated for 30 s at 100% humidity at 4°C in a Vitrobot mark IV, blotted for 3 s with blot-force 8 and plunged into liquid ethane. A total of 9,709 micrograph movies were collected on a CryoArm200 cryo-electron microscope (JEOL) equipped with a K2 direct electron detector (Gatan), in-column energy filter, and cold field-emission gun (low-flash interval 4 h). A total dose of 40 e$^-$/A$^2$ was fractionated over 40 frames at a defocus range of −1.2 to −2.7 μm using SerialEM (134) in a 5 × 5 multi-hole strategy as described (65).

### Cryo-EM image processing and model building

Pre-processing was carried out using WARP (61), followed by 2D and 3D classification and auto-refinement using Relion 4.0 (62). During pre-processing motion-correction, CTF estimation and particle picking was performed. The pixel size was binned to 1.50846 Å/pix and particles extracted with a box size of 190. Rough 2D classification followed by 3D classification using a reference of hPol I obtained after stringent 2D classification and 3D refinement yielded a reconstruction at an overall resolution of 4.09 Å. Further 3D classification was performed to investigate the occupancy and flexibility of the dimerization domain of RPA49/34 and the clamp/stalk region. Models for common subunits RPABC1-5 and the RPAC1/2 assembly were transferred from a hPol III reconstruction (21). Homology models of the hPol I subunits RPA1, RPA2, RPA49, RPA34, RPA12, and RPA43 were generated based on sequence and secondary structure alignments with the crystal structures of their *S. cerevisiae* counterparts (Supplemental Data 1) using the MODELLER software package (63). The models were adjusted in COOT (135) and real-space refined using Phenix (136). At later stages, released AlphaFold (64) models were used to guide chain-tracing in poorly resolved areas and specifically modelling of the dock II domain was supported by its AlphaFold prediction. A model of the stalk subunit RPA43 is included in some figures, but was not deposited because of poor or absent cryo-EM density resulting from flexibility.

### In situ protein docking

To investigate the protein–protein interactions between hPol I and Top2a, we used HADDOCK (92), AutoDock Vina (93), ZDOCK webserver (94), and PRISM webserver (95, 96). We focused on the RPA1 subunit of hPol I (PDB-ID 7OBB, chain A) and considered both states of Top2a (state I: PDB-ID 6ZY7, state II: PDB-ID 6ZY8).

#### HADDOCK

For docking with the software package HADDOCK (92), first all amino acids of the solvent-accessible surface area were identified using FreeSASA (137). The segment containing residues 1,060–1,155 of the RPA1 subunit (chain A) of the polymerase was defined as the active docking part. The surface of Top2a was defined as passive docking partner and thus completely sampled by the RPA1 subunit. Finally, the complete polymerase complex was aligned to the docked RPA1 subunit and only those docking results were considered, which do not exhibit any overlap with Top2a.

#### AutoDock Vina

The software package AutoDock Vina (93) was used for rigid docking of the key fragment (residues 1,060–1,155) of the RPA1 subunit to Top2a in both states. Finally, the complete hPol I complex was aligned to the docked RPA1 subunit, and only those docking results were considered which do not exhibit any overlap with Top2a.

#### Webserver docking

We used the webservers ZDOCK (94) and PRISM using the default settings. Because of limitations of the webservers, we did not dock the complete RPA1 subunit, but (1) the key fragment (residues 1,060–1,155) of the RPA1 subunit (PDB-ID 7OBB, chain A), (2) a complex of this fragment with subunit RPABC1 (PDB-ID 7OBB, chain E), and (3) the HMG box 5 of humane UBF (PDB ID: 2HDZ) to both states of Top2a. Finally, the complete hPol I was aligned to the docked RPA1 subunit, and only those docking results were considered, which do not exhibit any overlap with Top2a.

#### Protein–protein interaction analysis

The key inter-protein atomic interaction patterns were identified and analyzed using the MAXIMOBY (CHEOPS) contact matrix algorithm and the VMD plugin PyContact (97).

## Data Availability

The cryo-EM density of human Pol I was deposited in the Electron Microscopy Data Bank under accession code EMD-15135. Model coordinates were deposited with the Protein Data Bank under accession code 8A43. Further material can be obtained from the corresponding author upon reasonable request.

## Supplementary Information

## Acknowledgements

The authors especially thank Philip Gunkel for his contribution. We thank all past and present members of the Engel lab, Achim Griesenbeck, Colyn Crane-Robinson, Christophe Lotz, Marlene Vayssieres, Klaus Grasser, Herbert Tschochner, and Philipp Milkereit for help and discussion; Gerhard Lehmann and Nobert Eichner for IT support; Joost Zomerdijk for UBF-constructs, Volker

Cordes for the Hela P2 cell line; Remco Sprangers for shared cell culture; Dina Grohmann and the Archaea Center for fermentation; and Thomas Dresselhaus for access to fluorescence microscopes. This work was in part supported by the Emmy-Noether Programm (DFG grant no. EN 1204/1-1 to C Engel) of the German Research Council and Collaborative Research Center 960 (TP-A8 to C Engel).

## Author Contributions

JL Daiß: data curation, formal analysis, investigation, visualization, methodology, and writing—review and editing.
M Pilsl: investigation, visualization, and methodology.
K Straub: validation, investigation, visualization, and methodology.
A Bleckmann: investigation.
M Höcherl: investigation.
FB Heiss: investigation.
G Abascal-Palacios: investigation and visualization.
EP Ramsay: investigation and visualization.
K Tlučková: methodology.
J-C Mars: investigation and visualization.
T Fürtges: methodology.
A Bruckmann: investigation.
T Rudack: supervision and methodology.
C Bernecky: supervision and methodology.
V Lamour: investigation.
K Panov: validation, investigation, and visualization.
A Vannini: supervision and validation.
T Moss: formal analysis, supervision, validation, and visualization.
C Engel: conceptualization, formal analysis, supervision, funding acquisition, validation, investigation, visualization, and writing—original draft, review, and editing.

## Conflict of Interest Statement

The authors declare that they have no conflict of interest.

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
