## [Reviewer comments · Life Science Alliance]

Life Science Alliance

The human RNA polymerase I structure reveals an HMG-like docking domain specific to metazoans

Julia Daiß, Michael Pils, Kristina Straub, Andrea Bleckmann, Mona Höcherl, Florian Heiss, Guillermo Abascal-Palacios, Ewan Ramsay, Katarina Tlučková, Jean-Clement Mars, Torben Fürtges, Astrid Bruckmann, Till Rudack, Carrie Bernecky, Valérie Lamour, Konstantin Panov, Alessandro Vannini, Tom Moss, and Christoph Engel

DOI: <https://doi.org/10.26508/lsa.202201568>

Corresponding author(s): Christoph Engel, University of Regensburg

Review Timeline:

Submission Date:	2022-06-20
Editorial Decision:	2022-07-27
Revision Received:	2022-08-03
Accepted:	2022-08-09

Transaction Report:

Please note that the manuscript was reviewed at Review Commons and these reports were taken into account in the decision-making process at Life Science Alliance.

July 27, 2022

RE: Life Science Alliance Manuscript #LSA-2022-01568

Prof. Christoph Engel
University of Regensburg
Regensburg Center for Biochemistry
Universitaetsstr. 31
Regensburg, Bavaria 93053
Germany

Dear Dr. Engel,

Thank you for submitting your revised manuscript entitled "The human RNA polymerase I structure reveals an HMG-like transcription factor docking domain specific to metazoans". We would be happy to publish your paper in Life Science Alliance pending final revisions necessary to meet our formatting guidelines.

- please consult our manuscript preparation guidelines <https://www.life-science-alliance.org/manuscript-prep> and make sure your manuscript sections are in the correct order
- please add a Category for your manuscript in our system
- please add a Running Title in our system
- please add a Summary Blurb/Alternate Abstract in our system
- please upload your main and supplementary figures as single files; all figure legends should only appear in the main manuscript file
- please add a callout for each Figure file to your main manuscript text; please update your callouts for the Supplementary Figures as well
- please add the Twitter handle of your host institute/organization as well as your own or/and one of the authors in our system
- please add sizes next to the blots in Figure 5
- the file labeled "Supplemental Data 1" needs to be separated and labeled as Supplemental Figures, with corresponding Figure Legends, and callouts within the text

A. FINAL FILES:

B. MANUSCRIPT ORGANIZATION AND FORMATTING:

Sincerely,

Reviewer #2 (Comments to the Authors (Required)):

The authors have significantly improved the quality of the manuscript and have added important new features. They have addressed most of my concerns and have justified their positions in case of disagreement. The manuscript deserves publication in the shortest delays.

August 9, 2022

RE: Life Science Alliance Manuscript #LSA-2022-01568R

Prof. Christoph Engel
University of Regensburg
Regensburg Center for Biochemistry
Universitaetsstr. 31
Regensburg, Bavaria 93053
Germany

Dear Dr. Engel,

Thank you for submitting your Research Article entitled "The human RNA polymerase I structure reveals an HMG-like docking domain specific to metazoans". It is a pleasure to let you know that your manuscript is now accepted for publication in Life Science Alliance. Congratulations on this interesting work.

DISTRIBUTION OF MATERIALS:

Again, congratulations on a very nice paper. I hope you found the review process to be constructive and are pleased with how the manuscript was handled editorially. We look forward to future exciting submissions from your lab.

Sincerely,
